# Systematic design of cell membrane coating to improve tumor targeting of nanoparticles

Lizhi Liu [1], Dingyi Pan [2], Sheng Chen[3], Maria-Viola Martikainen [4], Anna Kårlund[5], Jing Ke[6], Herkko Pulkkinen[1], Hanna Ruhanen [7,8], Marjut Roponen [4], Reijo Käkelä[7,8], Wujun Xu [1] ✉, Jie Wang [9] ✉ & Vesa-Pekka Lehto [1] ✉

Cell membrane (CM) coating technology is increasingly being applied in nanomedicine, but the entire coating procedure including adsorption, rupture, and fusion is not completely understood. Previously, we showed that the majority of biomimetic nanoparticles (NPs) were only partially coated, but the mechanism underlying this partial coating remains unclear, which hinders the further improvement of the coating technique. Here, we show that partial coating is an intermediate state due to the adsorption of CM fragments or CM vesicles, the latter of which could eventually be ruptured under external force. Such partial coating is difficult to self-repair to achieve full coating due to the limited membrane fluidity. Building on our understanding of the detailed coating process, we develop a general approach for fixing the partial CM coating: external phospholipid is introduced as a helper to increase CM fluidity, promoting the final fusion of lipid patches. The NPs coated with this approach have a high ratio of full coating (~23%) and exhibit enhanced tumor targeting ability in comparison to the NPs coated traditionally (full coating ratio of ~6%). Our results provide a mechanistic basis for fixing partial CM coating towards enhancing tumor accumulation.

Cell membrane (CM) coating has emerged as a desirable surface modification strategy to endow nanoparticles (NPs) with excellent biological interface properties including homologous targeting, efficient drug delivery, immune evasion, and long circulation time[1–4]. A wide variety of cell types including cancer cells, stem cells, immune cells, red blood cells (RBCs), platelets and human lung fibroblasts have been used as membrane sources to coat functional NPs for use in bioimaging[5,6], cancer immunotherapy[7–11], detoxification[12,13], and inhibition of severe acute respiratory syndrome coronavirus 2 (SARS-CoV-2)[14–16]. Examples

include hybrid CM-coated semiconducting polymer nanoengager for efficient photothermal immunotherapy[17], erythrocyte membrane-cloaked nanogel for glioblastoma treatment[18], cancer CM-camouflaged arsenene nanosheets that actively target cancer cells and show long-term retention in the circulation[19] and human macrophage membrane-coated poly(lactic-co-glycolic acid) (PLGA) NPs capable of blocking SARS-CoV-2 infection[20]. Of those, repeat extrusion through polycarbonate track-etched (PCTE) membranes and sonication approaches are commonly used to fuse core NPs with CMs[21]. However,

[1]Department of Applied Physics, University of Eastern Finland, 70210 Kuopio, Finland. [2]State Key Laboratory of Fluid Power and Mechatronic Systems, Department of Engineering Mechanics, Zhejiang University, Hangzhou 310027, China. [3]Department of Biomedical Engineering, Yale University, New Haven, CT 06511, USA. [4]Department of Environmental and Biological Sciences, University of Eastern Finland, 70210 Kuopio, Finland. [5]Institute of Public Health and Clinical Nutrition, University of Eastern Finland, 70211 Kuopio, Finland. [6]Department of Chemistry, Boston College, Chestnut Hill, MA 02467, USA. [7]Molecular and Integrative Biosciences Research Programme, Faculty of Biological and Environmental Sciences, University of Helsinki, 00014 Helsinki, Finland. [8]Helsinki University Lipidomics Unit (HiLIPID), Helsinki Institute of Life Science (HiLIFE) and Biocenter Finland, 00014 Helsinki, Finland. [9]School of Pharmacy, Anhui Medical University, Hefei 230032, China. ✉e-mail: wujun.xu@uef.fi; 2020500067@ahmu.edu.cn; vesa-pekka.lehto@uef.fi

we recently demonstrated that NPs partially coated with CM were the dominant (>90%) species in the final fusion product[22], when they were subjected to mechanical forces imposed by mechanical extrusion or ultrasonication. Although such partially coated NPs exhibit some targeting ability in vitro, only about 40% of partially coated NPs are internalized by source cells according to our proposed aggregation mechanism[22]. Given these drawbacks, much research interest has focused on ways to increase the ratio of full CM coating to improve tumor targeting efficiency, especially as only ~0.7% of injected NPs were found to accumulate at the tumor site[23]. Disruption of the membrane structure by the application of external forces (e.g., extrusion and sonication) is thought to initiate the subsequent spontaneous formation of the CM coating with an integrated core shell structure[21]. However, little is known about the mechanisms underlying how and whether the process of original CM rupture and final fusion occur resulting in partial coating. A better understanding of this coating process is needed to design an efficient procedure for CM coating in nanomedicine.

Here, inspired by the supported lipid bilayer (LB) formation process, we set out to identify the mechanism by which the coating process (adsorption, rupture, and fusion of CM vesicles) is responsible for partial coating. Using a combination of computational modeling and experimental analyses, we find that limited CM fluidity leads to failure of fusion of adjacent CM patches, thereby resulting in partial coating. Specifically, the adsorbed membrane patches can result from either the original CM fragments or rupture of CM vesicles. Given the critical role of membrane fluidity in regulating final fusion, we fix partial coating by tuning the CM fluidity using external phospholipid. Importantly, in vitro and in vivo experiments reveal that fixed partial coating effectively enhance the internalization of biomimetic NPs and tumor targeting. These results provide in-depth mechanistic insights into the generation of partial coating during the extrusion process and have implications for rational design of CM functionalized biomimetic NPs.

## Results

### Comparison of CM coating and LB coating

At present, most CM-coated NPs are prepared using a well-reported top-down approach (Fig. 1a), in which CM-derived vesicles are first obtained by emptying cells and extruded through a PCTE membrane, followed by co-extrusion with core NPs. This biomimetic design was inspired by LB membranes supported on solid substrates (Fig. 1b), which are widely used as artificial model membranes for monitoring biological processes (e.g., immune response and cell adhesion)[24]. In contrast to the preparation of CM-coated NPs that require external forces (e.g., extrusion or sonication), LB-coated NPs form spontaneously by fusion of liposomes with core NPs through both electrostatic and van der Waals interactions[25]. This difference prompted us to explore whether LB-coated NPs could retain membrane integrity. For this purpose, models of CM-coated $SiO_2$ (CM-$SiO_2$) NPs and zwitterionic 1,2-dioleoyl-sn-glycero-3-phosphocholine (DOPC) LB-coated $SiO_2$ (LB-$SiO_2$) NPs were employed in accordance with work reported previously[22]. The CM fragments used for coating were derived from mouse colon carcinoma (CT26) cells and the size of core mesoporous $SiO_2$ NPs was approximately 70 nm (Supplementary Fig. 1). Dynamic light scattering (DLS) analysis revealed that the average hydrodynamic diameter of CM vesicles was 138.1 ± 0.9 nm and slightly larger than that of liposomes (99.9 ± 0.7 nm) (Fig. 1c), with zeta potentials of −28.6 ± 0.8 mV and −1.4 ± 0.3 mV, respectively (Fig. 1d). Both CM vesicles and liposomes were spherical, homogeneous and unilamellar, as determined by cryogenic transmission electron microscopy (cryo-TEM) (Fig. 1e, f). Upon membrane coating, TEM images clearly revealed that the CM-$SiO_2$ NPs were partially coated, whereas the LB-$SiO_2$ NPs were fully coated when compared to the bare $SiO_2$ NPs (Fig. 1g–i). These observations correspond to the results of DLS, indicating that coating of $SiO_2$ NPs with CM or LB causes a consistent increase in the

hydrodynamic diameter of 10–20 nm and a change in the zeta potential (Fig. 1j). Finally, we calculated the ratios of full coating using our previously reported fluorescence quenching assay[22]. As expected, the ratio of full coating of LB-$SiO_2$ NPs (~54%) was much higher than that of CM-$SiO_2$ NPs (~6.3%; Fig. 1k), indicating that spontaneous deposition of liposomes on NPs favored full coating.

### Analysis of vesicle rupture

Based on the above quantitative results of the ratio of full coating, we next investigated why it was difficult to fully coat $SiO_2$ NPs with CM in comparison to LB-$SiO_2$ NPs. To address this issue, we first sought to clarify the mechanism underlying the formation of LB-$SiO_2$ NPs. The formation of LB-$SiO_2$ NPs involves three steps, as shown schematically in Supplementary Fig. 2a: (1) adsorption of liposomes onto the NP surface; (2) deformation of the liposomes and continued expansion in the contact area; (3) rupture of the deformed liposomes and generation of lipid patches, which eventually fuse with each other to form a full LB coating on the NP surface[26]. Motivated by this mechanism of LB-$SiO_2$ NPs formation, we focused on the adsorption process of CMs. Initially, the as-prepared CM materials consisted of both CM fragments and CM vesicles (Supplementary Fig. 3), suggesting two possible pathways to produce partial coating: direct adsorption of free CM fragments onto the surface of NPs; and adsorption and subsequent rupture of CM vesicles (Supplementary Fig. 2b). Traditionally, the latter pathway, involving fusion of CM vesicles with NPs, was thought to be the origin of CM coating. However, it is possible that the observed partial coating from TEM images was derived from the simple adsorption of free CM fragments onto the surface of NPs, as the TEM images only showed the final status after extrusion. Furthermore, CM vesicles contain extensive membrane proteins that may change their mechanical properties and could affect rupture behavior under the mechanical forces imposed by extrusion. To explore this possibility, we first performed liquid atomic force microscopy (AFM) to investigate differences in mechanical properties between liposomes and CM vesicles. AFM Young's modulus mapping revealed that the CM vesicles were stiffer than liposomes (Fig. 2a, b), in good agreement with earlier reports[27]. This difference was further confirmed by quantitative analysis of Young's modulus, with average stiffness values of 0.9 ± 0.3 MPa for liposomes and 3.6 ± 0.7 MPa for CM vesicles (Fig. 2c). These results regarding stiffness raise the question of whether rupture of CM vesicles could occur leading to partial coating.

Although extrusion devices are commonly used to produce CM-coated NPs, the actual mechanism by which the interaction between CM vesicles and NPs occurs during extrusion is not yet clear. To elucidate this process, PCTE membranes (pore size 200 nm) after extrusion were visualized by field emission scanning electron microscopy (FE-SEM) (Fig. 2d, e). We found that the pores were distributed randomly on the PCTE membrane and several aggregated NPs were located in the pores of the membrane (Fig. 2d). Surprisingly, the actual pore structure of PCTE membrane was non-parallel, resulting in the occurrence of pore channel crossover (Fig. 2e). This specific structure could contribute to the increased retention time of NPs and CM vesicles as well as the opportunity for interaction in the channel[28]. With CM vesicles larger than the PCTE membrane pore size of 200 nm for coating (i.e., 297.6 ± 6.9 nm), we found that the extrusion process was difficult because of significant blocking of the PCTE membrane by the CM (Supplementary Fig. 4), excluding the possibility of rupture on the surface of the PCTE membrane. Therefore, as the average size of CM vesicles (approximately 138 nm) actually used was smaller than the pore size (200 nm) and the length of the channel (8 μm) was much greater than the size of $SiO_2$ NPs (70 nm), we postulated that the entire coating process including adsorption, rupture and fusion occurred in the channels of the PCTE membrane (Fig. 2f).

It is not clear whether stiffer CM vesicles could be ruptured in the channels of the PCTE membrane, because this process has not been

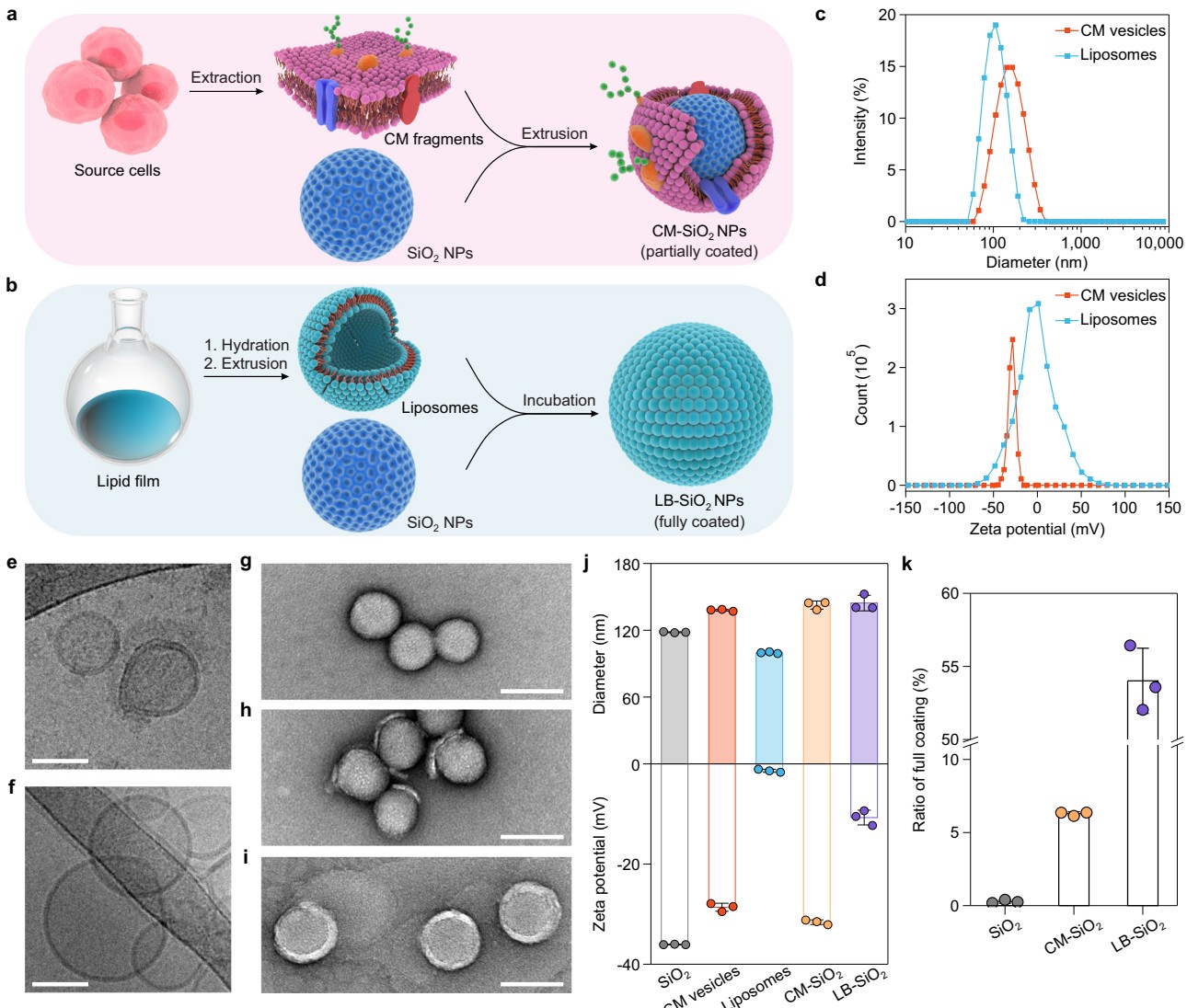

**Fig. 1 | Comparison of cell membrane (CM) coating and lipid bilayer (LB) coating. a** Schematic illustration of the preparation of CM-coated SiO$_2$ (CM-SiO$_2$) NPs through a physical co-extrusion method. **b** Scheme depicting the synthesis of LB-coated SiO$_2$ (LB-SiO$_2$) NPs. **c, d** Size distribution (**c**) and zeta potential (**d**) of CM vesicles and liposomes, as measured by DLS. **e, f** Cryo-TEM images of CM vesicles (**e**) and liposomes (**f**). Scale bars, 50 nm. **g–i** TEM images of bare SiO$_2$ NPs (**g**), CM- SiO$_2$ NPs (**h**), and LB-SiO$_2$ NPs (**i**). Scale bars, 100 nm. **j** Mean diameters and zeta potentials of SiO$_2$ NPs, CM vesicles, liposomes, CM-SiO$_2$ NPs, and LB-SiO$_2$ NPs. **k** Quantification of the ratio of full membrane coating for SiO$_2$ NPs, CM-SiO$_2$ NPs, and LB-SiO$_2$ NPs. Experiments in panels **c–i** were repeated three times independently with similar results. Data represent the mean ± s.d. (n = 3 independent experiments) in panels **j** and **k**. Source data are provided as a Source Data file.

directly observed in situ by SEM or TEM. We addressed this issue using a coupled Lattice Boltzmann−Phase Field model to simulate the adsorption and rupture of CM vesicles during extrusion (for more details, see the Model and Simulation section), due to the shear force of the flow in the microchannel is the main driving force for the rupture of CM vesicles. To determine the interaction of CM vesicles with core NPs in the channel, we simulated the case where the initial distance between NPs and CM vesicles was 0.2 μm, which is the same as the average diameter of the channel. Viewed from the distribution of the shear force along the surface of a CM vesicle (Fig. 2g) as well as the distribution of the inner and outer pressure difference along the surface of the CM vesicle (Fig. 2h), the pressure gradient field (∇p), which is the driving force for particle transport, was maximized at the edges of the CM vesicle, but remained almost constant elsewhere. This result suggests that the presence of the CM vesicle has little effect on the transportation of NPs in front of or behind it, but it can effectively become trapped in the microchannel. Consequently, the NP can slowly approach the CM vesicle within several τ driven by the shear force of

the flow. This observation was further confirmed by measurement of the minimal distance between the NP and CM vesicle (Fig. 2i). Conversely, we found that the NP and liposome could not attract each other under the same conditions (Supplementary Fig. 5), likely due to the different physical properties (e.g., size and stiffness) of liposomes and CM vesicles. These simulation results indicate that the CM vesicles can become attached to NPs in the channel during extrusion, which is a prerequisite for CM coating with the extrusion method.

Our next objective was to determine whether the deformation induced by the surface tension of the CM vesicles was sufficiently strong to rupture the membrane. During extrusion, CM vesicles experience high shear stress in streaming, which deforms their outline and increases membrane tension. Membranes rupture at relative expansion in the order of 2−4%, which corresponds to a typical surface tension value for the lysis threshold of -1 mN/m[29−31]. Hence, based on comparison of the maximum surface tension of deformed CM vesicles with this critical threshold, we could assess whether or not the CM vesicles would rupture. In our model, to better simulate the coupling

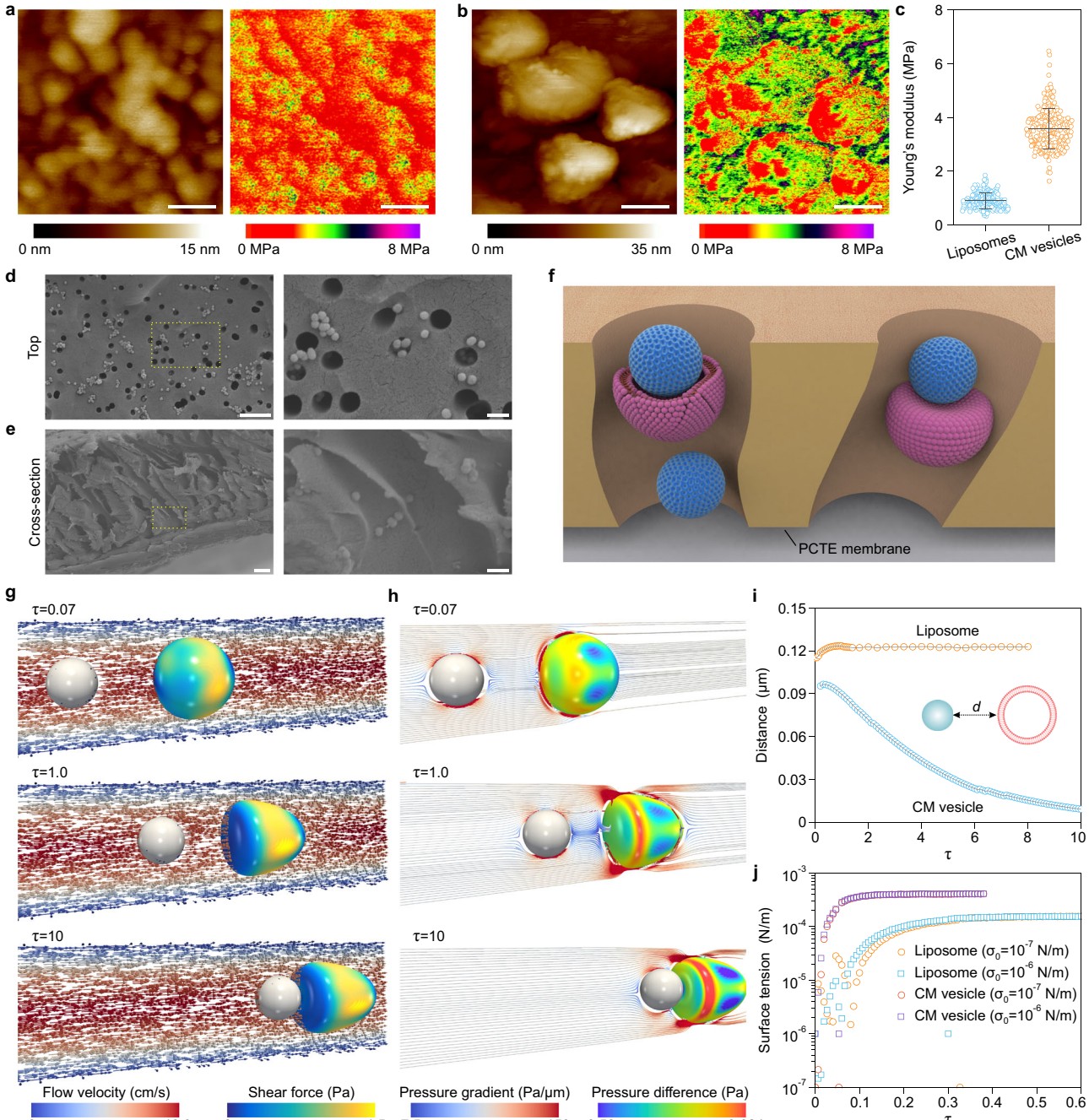

**Fig. 2 | Computational modeling of the interaction between CM vesicles and SiO₂ NPs during extrusion. a**, **b** AFM topographic images (left panel) of liposomes (**a**) and CM vesicles (**b**), along with the corresponding Derjaguin–Muller–Toporov Young's modulus map (right panel). Scale bars, 50 nm. **c** Quantification of the average Young's modulus of liposomes ($n = 212$ positions from 10 independent liposomes) and CM vesicles ($n = 229$ positions from 10 independent CM vesicles). Data represent the mean ± s.d. **d**, **e** SEM images of polycarbonate track-etched (PCTE) membrane after extrusion in top view (**d**) and cross-sectional view (**e**). Scale bars, 1 μm in low-magnification (left panel) and 200 nm in high-magnification (right panel). **f** Schematic illustration of the interaction between CM vesicles and SiO₂ NPs in the pore of PCTE membrane during extrusion. **g**, **h** Time evolution of CM vesicle deformation and positions with times taken as: $\tau = 0.07$, $\tau = 1.0$, and $\tau = 10$. **g** Distribution of the shear force along the surface of CM vesicle with the

background arrow field reflecting the surrounding fluid flow field. **h** Distribution of the inner and outer pressure difference along the surface of CM vesicle with the background field lines reflecting the pressure gradient ($\nabla p$) of the flow. Lines are tangent to $\nabla p$ at each point along the length and shrunk at the edges of the CM vesicle. Colors indicate the magnitude of $\nabla p$ (e.g., $|\nabla p|$). **i** Time evolution of the minimal distance between NP and CM vesicle/liposome during extrusion with the initial distance of 0.2 μm. The inset represents the distance ($d$) between the edge of NP and CM vesicle/liposome. **j** The surface tension as a function of dimensionless time $\tau$ ($\tau = t/t_0$, with $t_0 = 0.5 D_{channel}/v_{inlet}$, where $D_{channel}$ is the average diameter of the channel and $v_{inlet}$ is the average velocity of the inlet flow). Two initial surface tensions ($\sigma_0$; $10^{-6}$ and $10^{-7}$ N/m) for CM vesicles and liposomes were simulated. Experiments in panels **a**–**e**, were repeated three times independently with similar results. Source data are provided as a Source Data file.

among surface tension, deformation of vesicles and the bending elasticity of the LBs, the surface tension ($\sigma$) of CM vesicles in the Cahn-Hilliard equation was coupled with the increase in the membrane surface area ($\triangle S$) through the following modified Helfrich equation[30–32]:

$$\frac{\triangle S}{S_0} = \frac{k_B T}{8\pi\kappa} ln\left(\frac{\sigma}{\sigma_0}\right) + \frac{\sigma - \sigma_0}{K_a} \qquad (1)$$

where $S_0$ is the original vesicle surface area, $k_B$ is the Boltzmann constant, $T$ is the absolute temperature, $\sigma_0$ is the surface tension of the undeformed CM vesicle, $\kappa$ is the bending rigidity and $K_a$ is the area expansion modulus of the membrane. Along with extrusion, fluid shear stress produced remarkable deformation of the CM vesicle (Fig. 2g) and a change in local curvature (Fig. 2h), suggesting that the resulting surface tension may lead to rupture of the CM vesicles. To further validate our observations, we performed simulations without the addition of NPs and tested various initial surface tensions to monitor the changes in surface tension during extrusion. Our results revealed that the final surface tension for both CM vesicles and liposomes increased by more than 100-fold under pressure-driven flow inside the microchannel (Fig. 2j). Notably, in comparison with liposomes, CM vesicles with greater stiffness and larger size were prone to rupture as the maximum surface tension was close to the rupture tension (1 mN/m) and the relative expansion exceeded 6% (Supplementary Fig. 6). Therefore, the rupture of CM vesicles was expected to occur in the microchannel, indicating that in addition to free CM fragments, the membrane fragments produced by CM vesicles rupture could be responsible for partial coating.

## Mechanism of fixing of partial CM coating

Having excluded the possibility that partial coating results from the failure of vesicle rupture, we next analyzed the final fusion process. We began by calculating the amount of lipid used for coating, because theoretically the amount of lipid required for coverage of the NPs with a single bilayer should be equivalent to the surface area of the core NPs[33]. However, with regard to the area of CM coating, the protein weight of the CM is usually used to quantify the amount need for coating, ignoring the contribution of membrane-associated phospholipids. Therefore, we used a bicinchoninic acid (BCA) assay kit and phospholipid quantification assay kit to calculate the percentage of phospholipid in the composition of the collected CM fragments. The two most commonly used methods for isolating CM fragments were tested: homogenization with a Dounce homogenizer; and a membrane protein extraction kit (Fig. 3a). Intriguingly, the membrane protein extraction kit (48.04%) obtained more phospholipids than the Dounce homogenizer (29.87%). Comparison of the practical amount of phospholipid used for coating with the theoretical value (with DOPC as a reference) revealed that the amount of phospholipid prepared by the Dounce homogenizer (0.426 mg) and the membrane protein extraction kit (0.924 mg) were much higher than the theoretical value (0.32 mg) required for full coating of 1 mg of mesoporous SiO$_2$ NPs (Fig. 3b; details regarding the calculation are presented in Supplementary Note 1), suggesting that there was indeed enough lipid to form a single bilayer around the NP. Of note, although we used CM fragments obtained with the membrane protein extraction kit that contained a greater amount of phospholipid, the ratio of full coating did not increase (approximately 6%; data not shown). Because our results of CM composition could not explain the generation of partial coating, we decided to investigate other mechanisms of partial CM coating.

A previous study suggested that the presence of lipid patches (Fig. 3c) could induce the formation of a cylindrical rim at the edge of the membrane to avoid exposure of their hydrophobic chains to the aqueous environment[34]. Compared with the unperturbed bilayer, the recombinant molecular packing of these lipids generates an excess free energy, called line tension ($\Upsilon$), which is the main driving force for the fusion of adjacent lipid patches (Fig. 3d). Because pore closure is correlated with membrane viscosity (Fig. 3e; for more details, see Supplementary Note 2), we hypothesized that the membrane fluidity may be responsible for the final fusion process. To examine this possibility, we first measured the gel–liquid crystalline phase transition temperature ($T_m$) of CM vesicles and CM liposomes by differential scanning calorimetry (DSC) (Fig. 3f), as an indicator of membrane fluidity. The CM liposomes were made of lipids extracted from the CM without incorporation of membrane proteins. Unlike pure 1,2-dipalmitoyl-sn-glycero-3-phosphocholine (DPPC) liposomes with a pre-transition peak at about 36 °C and a main transition peak at 40.7 °C, both CM vesicles and CM liposomes exhibited several transition peaks from 20 °C to 60 °C due to the complex lipid components of the CM. These results indicate that not all of the lipids in the CM were in the liquid disordered phase at room temperature (i.e., 25 °C).

Intrigued by this finding, we next used two molecular probes, i.e., laurdan and 1,6-diphenyl-1,3,5-hexatriene (DPH), to further evaluate the membrane fluidity of CM vesicles at different temperatures. Functionally, laurdan is a polarity-sensitive fluorescent probe that can be used to assess membrane bilayer phase properties. For example, a low degree of water penetration in the membrane causes an increase in the generalized polarization (GP) of laurdan and indicates tighter lipid packing in the lipid headgroup region (decrease in membrane fluidity)[35]. DPH is a hydrophobic fluorescent probe that is commonly used to study the microviscosity of the bilayer interior: higher anisotropy values of DPH reflect restricted motion of the probe and thereby a more ordered membrane[36]. For pure DPPC liposomes, a drop in laurdan GP value from 0.2 to 0.1 was detected at 45 °C (Fig. 3g) as a consequence of the increase in membrane fluidity due to the phase transition of the phospholipid from the gel to liquid-crystalline phase, consistent with the DSC results (Fig. 3f). In the liquid-disordered phase, the laurdan GP value decreased monotonically with continually increasing temperature. In contrast, we found that the laurdan GP values of CM vesicles and CM liposomes decreased progressively with increasing temperature due to their multiple transition temperatures (Fig. 3g). Remarkably, compared with the CM liposomes, the presence of membrane proteins in the CM vesicles suppressed the membrane fluidity, probably due to restricted swing of CM lipids around membrane proteins[37]. The temperature dependencies of membrane fluidity were further confirmed by measuring the anisotropy values of DPH (Fig. 3h) and its corresponding microviscosity (Supplementary Fig. 7).

To better determine the functional role of lipid composition in membrane fluidity, we next identified and quantified the lipids in CMs by electrospray ionization mass spectrometry (ESI-MS), and compared extraction by the Dounce homogenizer and the membrane protein extraction kit (Fig. 3i and Supplementary Fig. 8). Lipidomics analysis revealed that the lipid compositions of the CMs collected by these two methods were similar, with only slight differences in the percentages of each lipid. Importantly, as the $T_m$ of phospholipids raised with decreasing unsaturation, the existence of such saturated phosphatidylcholine (PC) species, e.g., PC 30:0 and PC 32:0, could decrease the membrane fluidity of CM vesicles. Nonetheless, the CM vesicles exhibited a certain degree of membrane fluidity at room temperature because of the presence of unsaturated phospholipids species. Based on these results, we concluded that the membrane fluidity of CM vesicles at room temperature is limited by the specific lipids with saturated acyl chains and incorporation of extensive membrane proteins, and that such membrane viscosity acts on the line tension and results in failure to fix partial CM coating.

To provide direct experimental evidence of the proposed mechanism of fixing partial CM coating, we examined the ratio of full coating at three representative coating temperatures: 25, 37, and 60 °C (Fig. 3j). As expected, the ratio of full coating at 60 °C was significantly

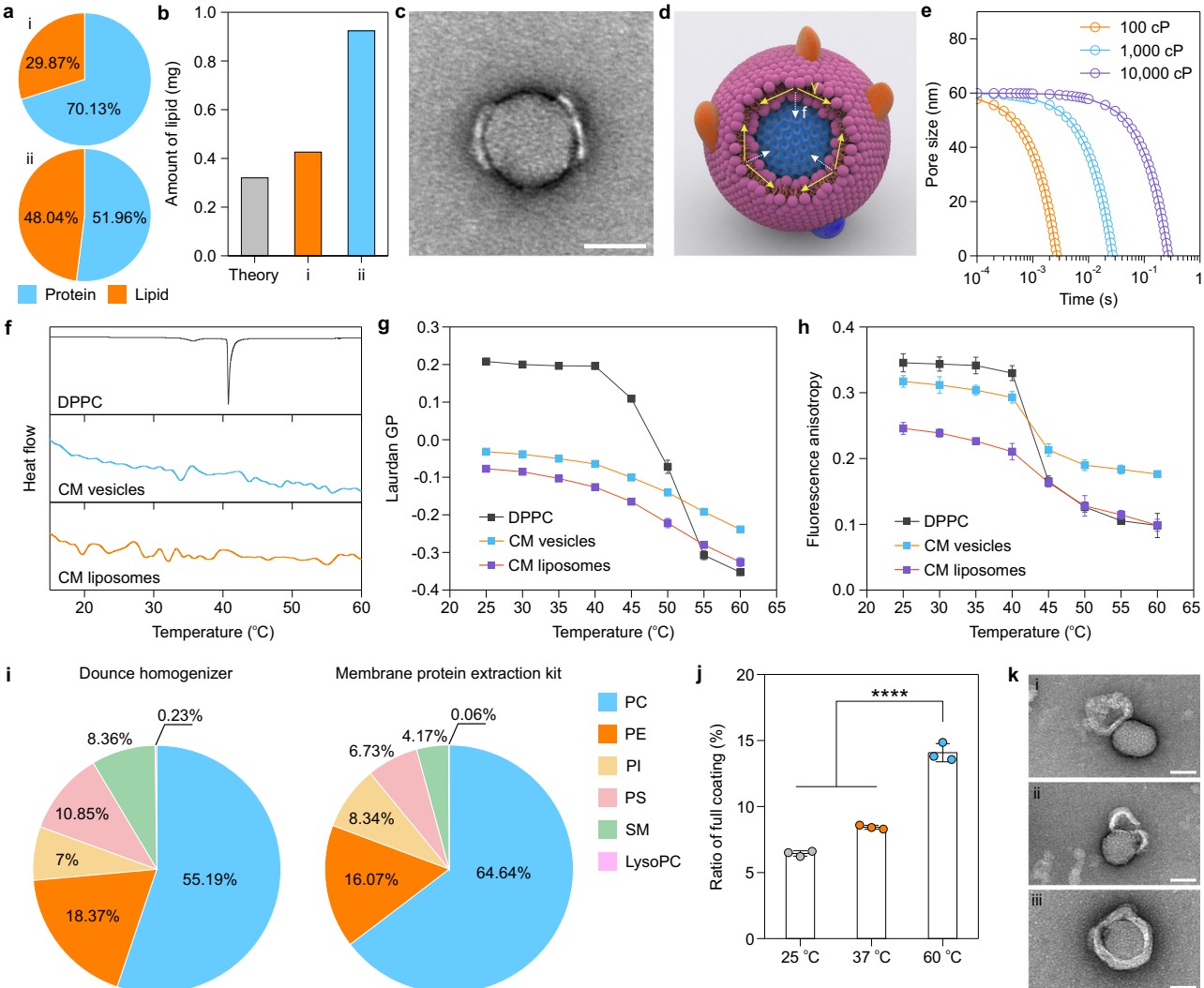

**Fig. 3 | Fixing of partial CM coating. a** Pie charts representing the weight percentage of membrane proteins and membrane-associated phospholipids in the collected CMs, where CM isolated by Dounce homogenizer (i) and membrane protein extraction kit (ii) were compared. **b** Comparison of the amount of lipid used for coating with the theoretical calculation of required lipid amount for full coating of NPs (1 mg) and experimental condition (i: Dounce homogenizer; ii: membrane protein extraction kit). **c** A typical TEM image of partially coated CM-SiO₂ NPs showing the two CM patches that need to be further fused. Scale bar, 50 nm. **d** Schematic illustration of the fixing of partial CM coating driven by the line tension (Υ). f: fixing force. **e** Pore size as a function of time and membrane viscosity (100, 1000 and 10,000 cP) under the constraint of membrane tension. **f** Differential scanning calorimetry (DSC) curves of DPPC liposomes, CM vesicles, and CM liposomes. **g, h** Temperature dependence of laurdan generalized polarization (GP; **g**) and fluorescence anisotropy (**h**) values in DPPC liposomes, CM vesicles, and CM

liposomes. **i** Comparison of plasma membrane composition obtained by extraction with the Dounce homogenizer and membrane protein extraction kit. Pie charts showing the mole percentage for each lipid class, as determined by mass spectrometry-based lipidomics profiling. The detailed species composition in each lipid class is presented in Supplementary Fig. 8. Phosphatidylcholine (PC), phosphatidylethanolamine (PE), phosphatidylinositol (PI), phosphatidylserine (PS), sphingomyelin (SM), and lysophosphatidylcholine (LysoPC). **j** Quantification of the ratio of full CM coating with different coating temperatures (25, 37, and 60 °C). **k** TEM images representing the different states of fixing of partial CM coating at 60 °C: adsorption (i), rupture (ii), and fusion (iii). Scale bars, 50 nm. Experiments in panels **a, c, f, i**, and **k**, were repeated three times independently with similar results. Data represent the mean ± s.d. (*n* = 3 independent experiments) in panels **g, h**, and **j**. Significance was determined by one-way ANOVA followed by post hoc Tukey test. ****\**p* < 0.001. Source data are provided as a Source Data file.

higher than those at 25 °C and 37 °C, indicating that an increase in membrane fluidity can facilitate the formation of full coating. This phenomenon was further supported by using two liposomes (DOPC and DPPC) with different membrane fluidity to coat SiO₂ NPs (Supplementary Fig. 9), which showed a decrease in the ratio of full membrane coating with increasing membrane stiffness. Hence, with increasing membrane fluidity, all procedures required for full CM coating including adsorption, rupture and fusion could occur in a similar manner to LB-SiO₂ NPs (Fig. 3k). Together, these observations indicate that membrane fluidity is directly related to the final fusion process for full CM coating.

## Construction and characterization of fixed HM-SiO₂ NPs

Encouraged by the above findings, we next sought to fix partial CM coating by modulating the membrane fluidity of CM vesicles. Although increasing the coating temperature is a potential strategy to improve membrane fluidity, high temperatures (e.g., 60 °C) would lead to irreversible loss of activity of membrane proteins, thereby limiting biomedical applications. To this end, we designed hybrid membrane (HM) vesicles to improve partial coating: these were composed of DOPC liposomes incorporating CT26 CM (Fig. 4a). We selected DOPC as a building block to fuse the CM due to its low T_m (−20 °C)[38], which could help reduce the T_m of CM vesicles below room temperature as

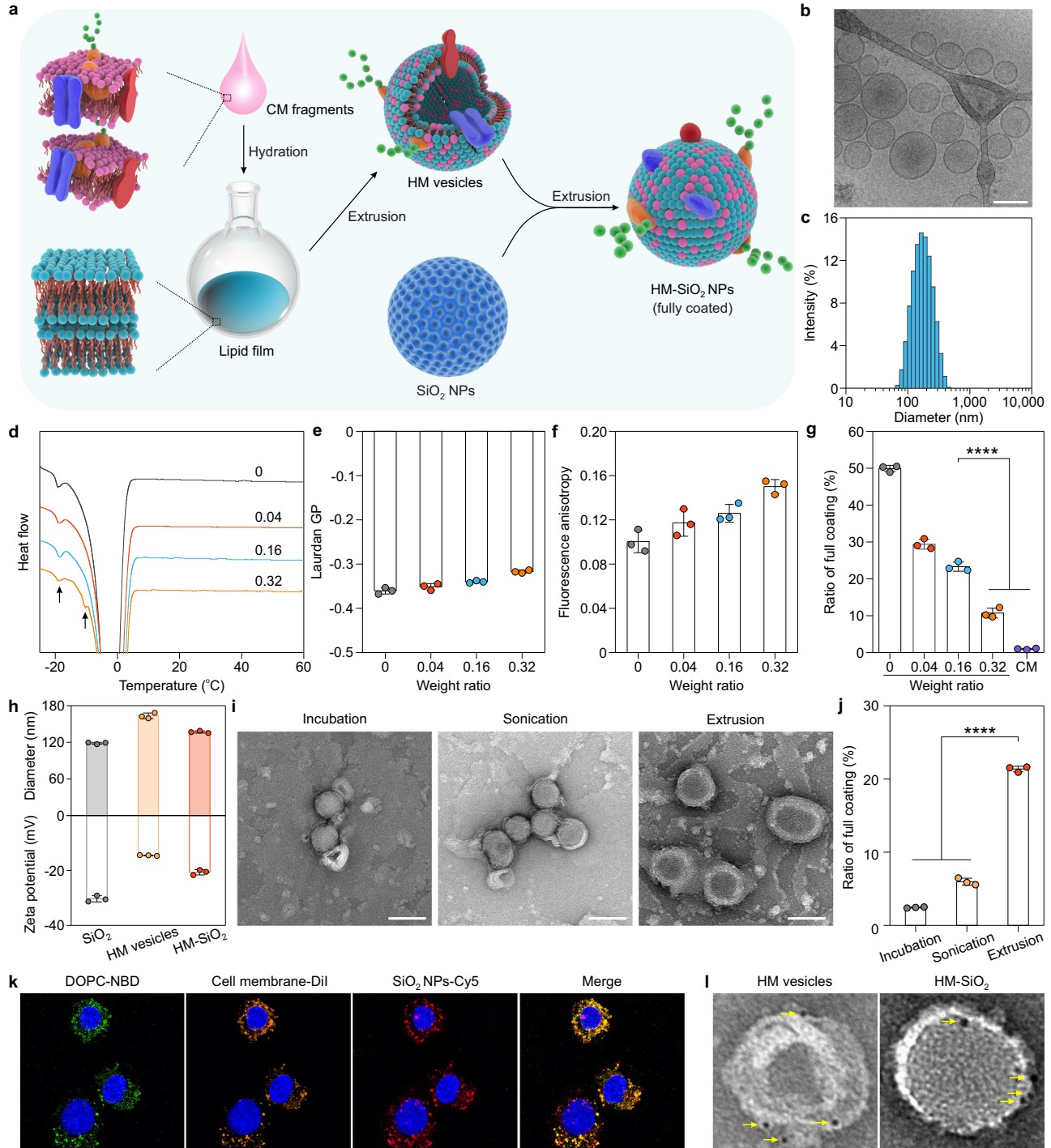

**Fig. 4 | Preparation and characterization of HM-SiO$_2$ NPs. a** Schematic showing the process of developing fully coated HM-SiO$_2$ NPs. **b** Cryo-TEM image of HM vesicles. Scale bar, 100 nm. **c** Size distribution of HM vesicles. **d** DSC analysis of HM vesicles prepared with CM protein/DOPC lipid weight ratios of 0, 0.04, 0.16, and 0.32. Typical peaks are indicated by black arrows. **e, f** Effects of CM incorporation on the laurdan GP (**e**) and fluorescence anisotropy (**f**) values of HM vesicles prepared at different weight ratios (protein/lipid). **g** Quantification of the ratio of full CM coating for SiO$_2$ NPs coated with HM vesicles of different weight ratios (protein/lipid) and the plain CM coating with the same amount of CM protein (0.4 mg/mL) used in the HM coating. **h** Mean diameters and zeta potentials of SiO$_2$ NPs, HM vesicles, and HM-SiO$_2$ NPs. **i** TEM images of HM-SiO$_2$ NPs fabricated by incubation, sonication, and extrusion. Scale bars, 100 nm. **j** Quantification of the ratio of full CM

coating with different coating methods (incubation, sonication, and extrusion). **k** CLSM images showing the intracellular co-localization of lipid (labeled with NBD; green), CM materials (labeled with DiI; orange) and SiO$_2$ NPs (labeled with Cy5; red) after internalization by CT26 cells. HM-SiO$_2$ NPs were incubated with CT26 cells for 4 h. Cell nuclei were stained with 4′,6-diamidino-2-phenylindole (DAPI) (blue). Scale bar, 10 μm. **l** TEM images of HM vesicles and HM-SiO$_2$ NPs stained with colloidal gold NPs (yellow arrows, ~5 nm). Scale bars, 25 nm. Experiments in panels **b**–**d**, **i**, **k**, and **l** were repeated three times independently with similar results. Data represent the mean ± s.d. ($n = 3$ independent experiments) in panels **e**–**h** and **j**. Significance was determined by one-way ANOVA followed by post hoc Tukey test in panels **g** and **j**. ****$p < 0.0001$. Source data are provided as a Source Data file.

well as avoid the formation of CM aggregates. After extrusion with a 200 nm PCTE membrane filter, the HM vesicles exhibited a spherical morphology and a unilamellar structure (Fig. 4b) with a homogeneous size of roughly 160 nm, as demonstrated by DLS (Fig. 4c), highlighting the functional role of DOPC in integration with external CMs to form integrated vesicles. To determine hybridization between the DOPC membrane and CM, a recently reported fluorescence colocalization method was employed[39–41]. CT26 cells were treated with HM vesicles for 4 h: DOPC was incorporated with 2% 1,2-distearoyl-$sn$-glycero-3-phosphoethanolamine-$N$-(7-nitro-2-1,3-benzoxadiazol-4-yl) (NBD-DSPE, green) and the CM materials was labeled with 1,1′-dioctadecyl-3,3,3′,3′-tetramethylindocarbocyanine perchlorate (DiI, orange). Confocal laser scanning microscopy (CLSM) images revealed that the green fluorescence derived from lipids matched well with the orange fluorescence derived from the CT26 CM (Supplementary Fig. 10a), which is supplementary evidence to indicate the successful co-fusion of the DOPC lipid membrane and CM. Moreover, Fourier transform infrared (FTIR) analysis revealed similar typical protein absorption bands (1500–1700 cm$^{-1}$) in the CM and HM vesicles relative to those found in the DOPC membrane (Supplementary Fig. 10b), further verifying successful hybridization.

To determine whether the incorporation of the CM influenced the membrane fluidity of HM vesicles, we evaluated HM vesicles prepared with three different weight ratios of CM protein to DOPC: 0.04, 0.16, and 0.32. In comparison with the control DOPC liposomes ($T_m$ = −19.2 °C), the ratio of 0.16 ($T_m$ = −15.97 °C) had the greatest incorporation of CM fragments, followed by 0.04 ($T_m$ = −16.16 °C) and 0.32 ($T_m$ = −19.03 °C; Fig. 4d). The observation that the degree of CM fragment integration within the LB was associated with increases in $T_m$ could be attributed to the packing effect of CM proteins[42]. Notably, at the highest ratio (0.32), we observed a similar $T_m$ to the control DOPC liposomes and an emerging $T_m$ (−9.29 °C), indicating that there is a threshold above which the HM vesicles cannot be further inserted with membrane proteins. Additionally, measurements of both laurdan GP (Fig. 4e) and microviscosity (Fig. 4f and Supplementary Fig. 11) revealed that the ratio of 0.16 was associated with a slight decrease in membrane fluidity compared with the control DOPC liposomes, but was still better than the original CM vesicles (Fig. 3g, h and Supplementary Fig. 7). Finally, when we applied these HM vesicles coated onto SiO$_2$ NPs through extrusion, we found that the ratio of full coating for the ratio of 0.16 was approximately 23.3%, which was slightly lower than for the ratio of 0.04 (~29.4%) but significantly higher than for 0.32 (~10.7%) and the plain CM coating (~1.0%; Fig. 4g). Therefore, we used a 0.16 mass ratio of CM protein to DOPC for subsequent experiments based on the compromise between amount of protein, membrane fluidity and final full coating ratio. Additionally, the HM coating method can be extended to other core NPs such as gold (Au) NPs and poly(lactic-co-glycolic acid) (PLGA) NPs (Supplementary Fig. 12) to improve the traditional CM coating. Besides the monounsaturated DOPC, the polyunsaturated 1,2-dilinolenoyl-sn-glycero-3-phosphocholine (18:3 PC) was used as a helper phospholipid to further validate the HM coating method (Supplementary Fig. 13) due to its higher membrane fluidity as compared to the monounsaturated DOPC[43]. The polyunsaturated 18:3 PC used in the HM coating gave a higher ratio of full coating (~34.5%) than that coated with monounsaturated DOPC (~23.3%). This finding further supports the conclusion that the high membrane fluidity is crucial for improving the traditional CM coating. Together, these findings indicate that the introduction of helper lipid can improve the membrane fluidity of original CM vesicles, which could favor the improvement of CM coating.

We next focused on the physicochemical characterization of fixed HM-SiO$_2$ NPs. According to DLS analyses (Fig. 4h), fusion SiO$_2$ NPs with HM vesicles caused a subtle increase in the hydrodynamic diameter of the SiO$_2$ NPs from ~118 to ~137 nm and a change in zeta potential from −30.3 to −20.7 mV. TEM images revealed that the extrusion process more effectively formed a full CM coating than incubation and sonication treatment (Fig. 4i), which is consistent with our previous report[22] and was further confirmed by quantification of the ratio of full coating (Fig. 4j). In stability studies, the morphology, size distribution and ratio of full coating of HM-SiO$_2$ NPs changed minimally before and after lyophilization (Supplementary Fig. 14), suggesting good potential for long-term storage and convenient for transportation. Furthermore, the fixed HM-SiO$_2$ NPs had good colloidal stability when suspended in phosphate-buffered saline (PBS) at 37 °C for a period long than 2 days (Supplementary Fig. 15a).

Finally, the integrative structure of HM-SiO$_2$ NPs was confirmed by fluorescence co-localization between DOPC (green), CM materials (orange) and SiO$_2$ NPs (red), which were marked by NBD, DiI, and Cyanine 5 (Cy5), respectively. The CLSM images revealed that with the process of serial extrusions, the green, orange, and red fluorescence signals in HM-SiO$_2$ NPs were nearly merged (Fig. 4k), indicating co-fusion of the HM vesicles and SiO$_2$ NPs. In addition to the reservation of lipids, the retention of membrane proteins on HM-SiO$_2$ NPs was verified by sodium dodecyl sulfate–polyacrylamide gel electrophoresis (SDS-PAGE) (Supplementary Fig. 15b). Compared with bare SiO$_2$ NPs and DOPC liposomes, HM vesicles and HM-SiO$_2$ NPs retained almost all of the original CM proteins. Moreover, colloidal gold staining (Supplementary Fig. 16) followed by TEM imaging provided direct visual evidence that CM proteins were successfully integrated into the HM vesicles and HM-SiO$_2$ NPs (Fig. 4l). Together, these results confirm that the HM-SiO$_2$ NPs inherited the key components from both DOPC and the CM.

## Cellular uptake of HM-SiO$_2$ NPs

To examine whether the fixed HM-SiO$_2$ NPs could exhibit homotypic targeting ability and improve cellular internalization efficiency, the HM-SiO$_2$ NPs as well as controls (SiO$_2$ NPs, LB-SiO$_2$ NPs, and CM-SiO$_2$ NPs) were incubated with different cell types including CT26 cells, HeLa human cervical carcinoma cells and MCF-7 human breast cancer cells (Fig. 5a–e). Initially, the biocompatibility of HM-SiO$_2$ NPs was tested on CT26 cells and no cytotoxicity was observed (Supplementary Fig. 17). CLSM revealed that CT26 cells had a higher red fluorescent signal (Cy5-labeled NP-based formulations) than the other cell lines after incubation with CM-SiO$_2$ NPs and HM-SiO$_2$ NPs, whereas all of the cell lines treated with SiO$_2$ NPs and LB-SiO$_2$ NPs had similar weak red fluorescence signals (Fig. 5a), verifying the targeting ability of the CM to its source tumor cells. More importantly, the CT26 cells incubated with HM-SiO$_2$ NPs had stronger fluorescent signals than those treated with CM-SiO$_2$ NPs, suggesting that the fixing of partial CM coating could improve targeting efficiency. The intracellular uptake by the different cell types was further measured quantitatively by flow cytometry (Fig. 5b–e). The results revealed that HM-SiO$_2$ NPs did not have affinity advantages for HeLa cells and MCF-7 cells, indicating a specific targeting ability of HM-SiO$_2$ NPs to homologous CT26 cells. Moreover, after incubation with NPs for 4 h, the fluorescence intensity of HM-SiO$_2$ NPs was significantly stronger than those of SiO$_2$ NPs, LB-SiO$_2$ NPs, and CM-SiO$_2$ NPs in CT26 cells. The increased cellular uptake of HM-SiO$_2$ NPs was further confirmed by TEM in CT26 cells (Supplementary Fig. 18), in which a large number of HM-SiO$_2$ NPs localized in the endocytic vesicles were observed in CT26 cells. We attributed the enhanced uptake efficiency of HM-SiO$_2$ NPs to two main reasons. First, despite a ratio of full coating of HM-SiO$_2$ NPs of approximately 23.3%, the percentage of HM-SiO$_2$ NPs with a coating degree larger than 50% was 82.3% (Supplementary Fig. 19a). This value was 11-fold larger than that of reported CM-SiO$_2$ NPs (7.4%)[22], indicating that most of the HM-SiO$_2$ NPs can individually enter the cells according to our proposed endocytic entry mechanism[22]. Second, due to the higher coating degree of HM-SiO$_2$ NPs, the HM-SiO$_2$ NPs had greater preservation of CM proteins than CM-SiO$_2$ NPs when we utilized equal masses of CM

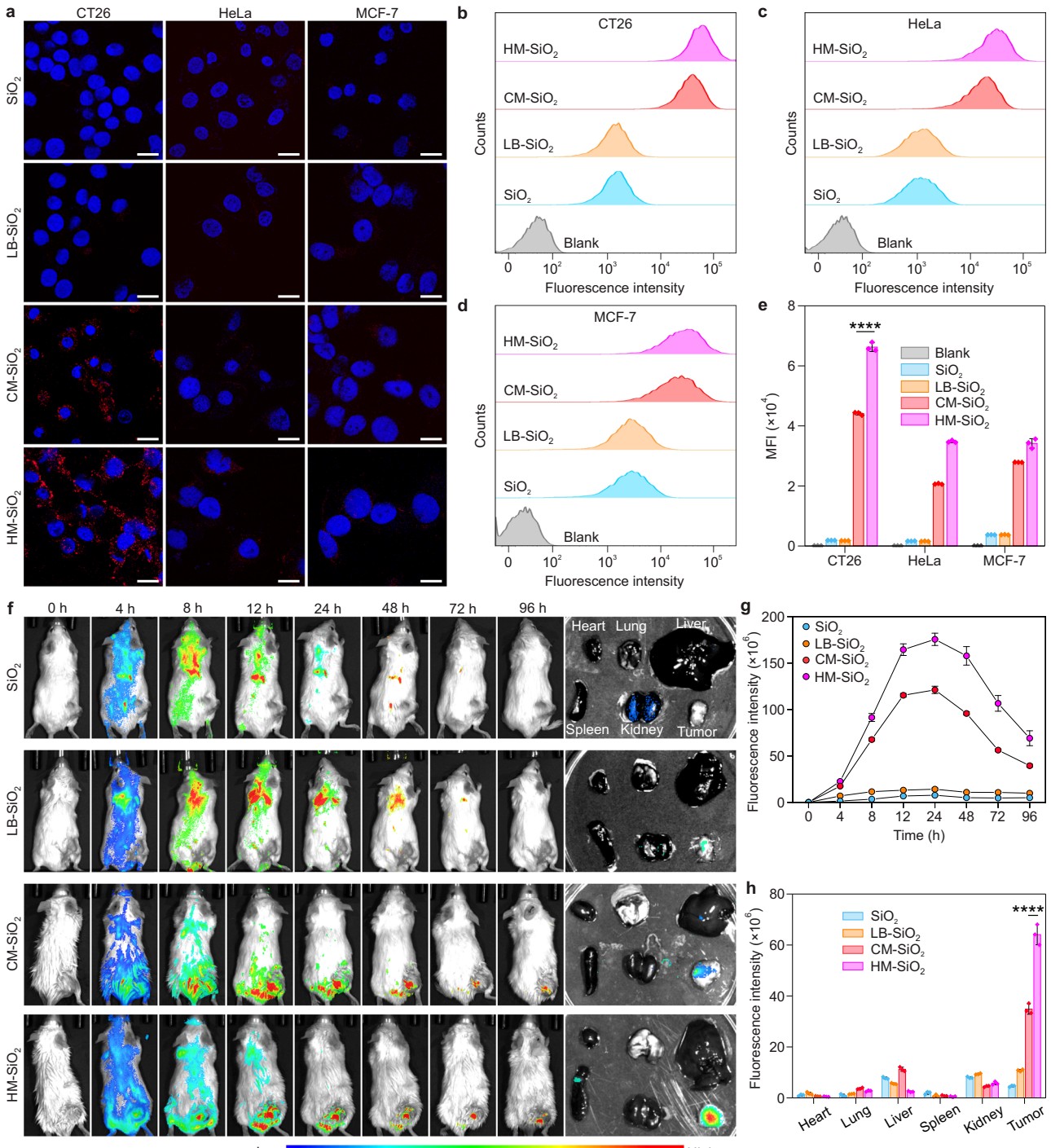

**Fig. 5 | Investigation of the homologous targeting capabilities of HM-SiO$_2$ NPs in vitro and tumor targeting in vivo. a** Typical CLSM images of three cancer cell lines (CT26, HeLa, and MCF-7) incubated with SiO$_2$ NPs, LB-SiO$_2$ NPs, CM-SiO$_2$ NPs, and HM-SiO$_2$ NPs for 4 h. Blue, cell nuclei stained with 4′,6-diamidino-2-pheny-lindole (DAPI); red, Cy5-labeled SiO$_2$ cores. Scale bars, 20 μm. **b**–**d** Flow cytometric analysis of CT26 cells (**b**), HeLa cells (**c**), and MCF-7 cells (**d**) after 4 h incubation with blank solution, SiO$_2$ NPs, LB-SiO$_2$ NPs, CM-SiO$_2$ NPs, and HM-SiO$_2$ NPs. **e** Quantification of the mean fluorescence intensities (MFI) for the three cell lines (CT26, HeLa, and MCF-7). Data represent the mean ± s.d. (*n* = 3 biologically inde-pendent cells). **f** Left panel: representative in vivo fluorescence images of CT26 tumor-bearing mice at different time points after intravenous injection of SiO$_2$ NPs,

LB-SiO$_2$ NPs, CM-SiO$_2$ NPs, and HM-SiO$_2$ NPs. Right panel: ex vivo fluorescence images of tumor and major organs (heart, lung, liver, spleen, and kidney) collected at 96 h after intravenous injection. **g** Time-dependent variation of fluorescence intensity at the tumor site after intravenous injection. **h** Quantitative region-of-interest (ROI) analysis of the fluorescent signals from the tumor and major organs collected at 96 h after intravenous injection. Experiments in panels **a**–**d** and **f** were repeated three times independently with similar results. Data represent the mean ± s.d. (*n* = 3 biologically independent mice) in panels **g** and **h**. Significance was determined by one-way ANOVA followed by post hoc Tukey test in panels **e** and **h**. ****$p < 0.0001$. Source data are provided as a Source Data file.

protein and SiO$_2$ NPs for coating (Supplementary Fig. 19b), indicating that the HM coating method had higher utilization efficiency of CM proteins than the traditional CM coating. Therefore, compared with CM-SiO$_2$ NPs, the HM-SiO$_2$ NP possessing a greater number of functional proteins per NP could provide sufficient receptor–ligand binding strength to drive NPs over the energy barrier during internalization.

Next, the immune evasion capability of HM-SiO$_2$ NPs was examined using the murine macrophage-like cell line, RAW264.7. Both CLSM (Supplementary Fig. 20a) and flow cytometry (Supplementary Fig. 20b, c) suggested that compared with bare SiO$_2$ NPs and LB-SiO$_2$ NPs, RAW264.7 cells had significantly reduced binding and/or internalization of CM-SiO$_2$ NPs and HM-SiO$_2$ NPs. Notably, flow cytometry revealed that the cellular uptake of HM-SiO$_2$ NPs in RAW264.7 cells was lower than that of CM-SiO$_2$ NPs, which was mainly because the full coating contained more CM proteins (Supplementary Fig. 19b). These results demonstrate that SiO$_2$ NPs coated with HMs also possess stealth ability to avoid clearance by the reticular endothelial system, which would be beneficial for tumor targeting in vivo.

### Tumor targeting capability of HM-SiO$_2$ NPs
Prior to testing the tumor targeting capability of HM-SiO$_2$ NPs, we determined the in vivo pharmacokinetic profiles of Cy5-labeled SiO$_2$ NPs, LB-SiO$_2$ NPs, CM-SiO$_2$ NPs, and HM-SiO$_2$ NPs to study whether the HM coating can prolong the blood circulation time (Supplementary Fig. 21 and Supplementary Table 3). According to the two-compartment model to fit the plasma concentration–time curves of these NPs, the bare SiO$_2$ NPs were rapidly cleared from the central compartment following first order processes with a distribution half-life ($T_{1/2\alpha}$) of $0.25 \pm 0.02$ h and an elimination half-life ($T_{1/2\beta}$) of $9.1 \pm 0.8$ h. The obtained $T_{1/2\beta}$ for the bare SiO$_2$ NPs is consistent with the values previously reported ca. 11 h for 160 nm SiO$_2$ NPs[44] and ca. 78 h for 33 nm SiO$_2$ NPs[45]. In contrast, the membrane-coated NPs exhibited a significantly slower clearance. Specifically, the $T_{1/2\beta}$ of HM-SiO$_2$ NPs ($23.6 \pm 2.3$ h) was longer than that of LB-SiO$_2$ NPs ($16.2 \pm 1.5$ h) and CM-SiO$_2$ NPs ($18.5 \pm 1.1$ h), suggesting that the fixing of partial CM coating was able to prolong the circulation time of the NPs in the blood.

To ascertain whether HM-SiO$_2$ NPs can selectively target tumor tissues, CT26 cells were implanted as subcutaneous xenografts into BALB/c mice. The tumor-bearing mice were injected intravenously with Cy5-labeled SiO$_2$ NPs, LB-SiO$_2$ NPs, CM-SiO$_2$ NPs, and HM-SiO$_2$ NPs. As shown in Fig. 5f, the fluorescence of SiO$_2$ NPs and LB-SiO$_2$ NPs was largely distributed in the lung and liver, but negligible fluorescence was detected at the tumor site. In contrast, mice injected with CM-SiO$_2$ NPs or HM-SiO$_2$ NPs had apparent fluorescence at the tumor sites from 8 h post-injection, most likely as a result of homotypic binding of the CMs to CT26 cells, indicating the accumulation of SiO$_2$ NPs at the tumor sites. Most importantly, mice treated with HM-SiO$_2$ NPs had much higher fluorescence intensity than those treated with CM-SiO$_2$ NPs, indicating the notable targeting capability of HM-SiO$_2$ NPs. We attributed this marked difference to enhanced immune escape and improved specific cancer targeting ability of fixed HM-SiO$_2$ NPs. Over time, HM-SiO$_2$ NPs continuously accumulated at the tumor site and the fluorescence intensity reached a maximum at 24 h. Relatively strong fluorescence was maintained for 96 h after injection, while fluorescent signals from other parts of the body were no longer observable (Fig. 5f, g). The mice were sacrificed at 96 h post-injection and tumors and other major organs were collected for ex vivo imaging. Consistent with the results of non-invasive imaging, the fluorescence intensity of the HM-SiO$_2$ NPs retained at tumor sites was significantly higher than those of other NPs groups (SiO$_2$ NPs, LB-SiO$_2$ NPs, and CM-SiO$_2$ NPs; Fig. 5f). Quantitative region-of-interest (ROI) analysis revealed that fluorescence intensity at the tumor sites of mice treated with the HM-SiO$_2$ NPs was approximately 14-, 6- and 2-fold higher than

those treated with SiO$_2$ NPs, LB-SiO$_2$ NPs, and CM-SiO$_2$ NPs, respectively (Fig. 5h). In addition, histological analysis of the major organs, including the heart, liver, lung, spleen and kidney, showed no pathological abnormalities in mice treated with SiO$_2$ NPs, LB-SiO$_2$ NPs, CM-SiO$_2$ NPs or HM-SiO$_2$ NPs (Supplementary Fig. 22), indicating that these NPs had negligible systemic toxicity. Collectively, these results suggest that the fixing of partial CM coating could effectively improve NPs enrichment within tumor tissue.

## Discussion
The cell membrane coated NPs with non-optimized techniques have shown promising biointerfacing properties. To further improve the coating techniques for getting more fully coated NPs, it is essential to understand the specific steps involved in the coating procedure and the parameters affecting it. With appropriate biomimetic, coating the functionality of the membrane can be exploited better leading to advances in the development of nanovectors and therapeutic agents. Computational simulations of interactions between CM vesicles and NPs during extrusion helped us to clarify how the CM vesicles attach to NPs and when (or whether) the rupture of CM vesicles occurs. Moreover, our simulation results of liposomes could potentially explain why it is better to use the incubation method to prepare the supported LB on NPs instead of the extrusion method. After demonstrating the possibility of CM vesicles rupture, a striking feature of our work was the discovery that the membrane fluidity is a regulator of fixing the partial coating, which in turn provides the rationale for targeted strategies to improve partial coating. Biointerfacing functions of CM-coated NPs are mainly due to surface modification of membrane proteins[46]. However, insertion of abundant membrane proteins into CM materials inevitably reduces the membrane fluidity, producing CM fragments and preventing the final fusion of membrane patches to achieve full coating. Here, this inherent contradictive behavior was mitigated by utilizing extrinsic lipids with low $T_m$, which improved membrane fluidity and facilitated to obtain fully coated NPs. The coating process may be further optimized via emerging technologies such as microfluidic sonication/electroporation[47,48] and flash nanocomplexation[49], but our developed HM approach for coating opens an avenue toward the improvement of partial CM coating to enhance tumor targeting.

In summary, by providing answers to two key questions regarding the rupture of CM vesicles and the fixing of partial CM coating, our findings substantially advance understanding of CM coating. Further construction of HM-SiO$_2$ NPs with higher ratios of full coating and enhanced tumor targeting efficiency in vitro and in vivo will provide new insights into the design of biomimetic NPs as well as active tumor targeting delivery.

## Methods
### Materials
Tetraethyl orthosilicate (TEOS), hexadecyltrimethylammonium bromide (CTAB), Au NPs (size: 80 nm), methyl-tert-butyl ether (MTBE), laurdan, 1,6-diphenyl-1,3,5-hexatriene (DPH), fetal bovine serum, antibiotic antimycotic solution (100×), trypsin-EDTA, 4′,6-diamidino-2-phenylindole (DAPI), 1,1′-dioctadecyl-3,3,3′,3′-tetramethylindocarbocyanine perchlorate (DiI), polycarbonate track-etched (PCTE) membranes (pore size: 0.1 μm, 0.2 μm, and 0.4 μm), and phospholipid quantification assay kit were purchased from Sigma-Aldrich Corporation (St Louis, MO, USA). Cyanine 5 (Cy5) was purchased from Lumiprobe Corporation (Hallandalae Beach, FL, USA). 1,2-dioleoyl-sn-glycero-3-phosphocholine (DOPC), 1,2-dipalmitoyl-sn-glycero-3-phosphocholine (DPPC), 1,2-dilinolenoyl-sn-glycero-3-phosphocholine (18:3 PC), and 1,2-distearoyl-sn-glycero-3-phosphoethanolamine-N-(7-nitro-2-1,3-benzoxadiazol-4-yl) (NBD-DSPE) were purchased from Avanti Polar Lipids Inc. (Alabaster, AL, USA). Bicinchoninic acid (BCA) assay kit, 6-(7-Nitrobenzofurazan-4-ylamino)dodecanoic acid NHS ester (NBD-X), and membrane protein

extraction kit were purchased from Thermo Fisher Scientific (Waltham, MA). 4-(2-hydroxyethyl)−1-piperazineethanesulfonic acid (HEPES), Dulbecco's modified Eagle's medium (DMEM), and RPMI 1640 medium were purchased from Biowest Corporation, France. Phosphate buffer saline (PBS) and Hank's balanced salt solution (HBSS) were obtained from HyClone. CellTiter-Glo assay was purchased from Promega Corporation, USA. Colloidal Au NPs (~5 nm)[50] and poly(lactic-co-glycolic acid) (PLGA) NPs[51] were synthesized according to the literatures. All chemical reagents were used directly without further purification unless specifically mentioned.

## Preparation of mesoporous SiO₂ NPs

To prepare mesoporous $SiO_2$ NPs, 6.24 g of CTAB, 0.3 g of sodium acetate trihydrate, and 53.4 mL of deionized water were continuously stirred at 60 °C for 2 h. Then, 4.35 mL of TEOS was added dropwise to the surfactant solution under vigorous stirring, and then heated at 60 °C for 12 h. After centrifugation, the products were washed with ethanol and transferred to a mixture of ethanol (100 mL) and concentrated hydrochloric acid (2.0 mL) with reflux and continual stirring at 90 °C for 24 h to remove CTAB. The as-synthesized NPs were washed three times with ethanol and resuspended in absolute ethanol for the storage.

## Synthesis of CM-SiO₂ NPs

To obtain CM-SiO₂ NPs, the membrane materials of CT26 cells were first extracted using the Dounce homogenizer according to the described method in our previous study[22]. For the comparison, the cell membranes (CMs) were also derived using the membrane protein extraction kit according to the manufacturer's standard protocol. Of note, in a typical experiment, unless stated otherwise, the CMs used for analysis or coating were obtained by using the Dounce homogenizer. The membrane-associated proteins and phospholipids in the isolated CMs were further analyzed and quantified. Among them, membrane proteins were quantified using a BCA assay kit and membrane-associated phospholipids were assayed using a phospholipid quantification assay kit.

To create CM-derived vesicles, the resulting CM materials were then physically extruded through a 200 nm PCTE membrane for at least 13 passes. Finally, the CM vesicles and SiO₂ NPs were mixed with the same mass (protein weight for CM vesicles) and co-extruded through a 200 nm PCTE membrane for at least 13 times to obtain CM-SiO₂ NPs. The excess free CMs were removed by centrifugation (6000 × g, 6 min).

## Preparation of LB-SiO₂ NPs

The DOPC lipid bilayer (LB) coated-SiO₂ (LB-SiO₂) NPs were prepared according to a procedure already described with minor modifications[52]. Briefly, 100 μL (25 mg/mL) of DOPC presolubilized in chloroform was added to the glass vials. For fluorescent labeling purpose, DOPC were mixed with a small fraction (2%) of NBD-DSPE. To obtain the lipid films, the chloroform in the vials was evaporated under a nitrogen flow in a fume hood. Then, the vials were placed in a vacuum desiccator at room temperature (25 °C) for 12 h to remove any residual chloroform. The lipid films were stored at −20 °C before use. To prepare liposomes, the lipid films were rehydrated in 1 mL of 20 mM HEPES buffer (200 mM NaCl, pH = 7.4) with occasional shaking for 1 h, forming a cloudy lipid suspension. Subsequently, the rehydrated lipid solution was extruded through a 100 nm PCTE membrane (minimum 31 cycles) using a mini-extruder purchased from Avanti Polar Lipids. The resulting uniform and unilamellar liposomes were stored at 4 °C.

The liposomes (2.5 mg/mL) and SiO₂ NPs were then mixed in equal volumes (usually 500 μL) and incubated at room temperature for 1 h with occasional agitation. The mixture was further purified by centrifugation to remove any excess lipids and finally dispersed in 1 mL of 20 mM HEPES buffer (pH = 7.4).

## Cryo-TEM imaging

The as-prepared liposomes (0.1 mg/mL), CM vesicles (0.5 mg/mL), and HM vesicles (0.1 mg/mL) were dropped on the holey carbon film-coated 200-mesh copper grids (Quantifoil Micro Tools GmbH) in the chamber of a FEI Vitrobot. The Vitrobot handled all of the operations, including blotting and plunge-freezing into liquid ethane. Imaging was performed on a JEM-3200FSC transmission electron microscope (JEOL, Tokyo, Japan) incorporating a liquid helium stage and an omega-type energy filter operating at 300 kV. During the imaging, the stage was cooled with liquid nitrogen to 80 K.

## Characterization of CM-SiO₂ NPs and LB-SiO₂ NPs

For the TEM characterization, as-synthesized NPs were placed on a TEM grid and left for 1 min, followed by washing with three drops of water. Then, the grids were negatively stained with 2% uranyl acetate (pH = 4.4) for 30 s. Excess solution was wiped away with absorbent paper and the samples were visualized using a JEM-2100F (JEM Ltd., Japan) microscope. The size distribution and zeta potential were measured using a dynamic light scattering (DLS; Malvern Zetasizer Nano ZS).

## Ratio of full CM coating measurement

The calculation of ratio of full CM coating was performed according to our previously reported fluorescence quenching assay[22]. Briefly, the NBD-labeled SiO₂ NPs as core materials for coating were first prepared through a sequential chemical surface modification. The fluorescence intensity was measured using a Synergy H1 microplate reader (Biotek, Winooski, USA; $\lambda_{exc}$ = 465 nm) after adding 20 μL of 1 mol/L sodium dithionite solution to the samples (100 μL). The ratio of full coating was determined using the following equation:

$$\text{Ratio of full coating}(\%) = \left[ (F_D - F_0)/(F_T - F_0) \right] \times 100 \quad (2)$$

where $F_T$ represents the total fluorescence of the sample without the addition of dithionite, $F_D$ represents the fluorescence of the sample after the addition of dithionite, and $F_0$ represents the background fluorescence.

## AFM imaging

The glass slides were first cleaned with 3% HCl in 96% ethanol for 1 h. Subsequently, they were coated with poly-l-lysine, followed by rinsed with deionized water, and dried overnight at room temperature (25 °C). The solution of CM vesicles and liposomes (50 μL) was dropped on the poly-l-lysine coated glass slide for the adhesion of samples, and gently rinsed with deionized water before imaging. AFM measurements were performed in liquid under the PeakForce mode using a Bruker Dimension FastScan AFM[53]. The spring constant of the used cantilevers was 0.3 N/m, and the standard AFM probe had a radius of 20 nm. To measure Young's modulus, force-displacement curves were obtained at 0.8 Hz. Young's modulus was estimated by converting the force curves into force-indentation curves and fitting them with the Derjaguin-Muller-Toporov model. Images were analyzed using the NanoScope analysis 1.9 version software (Bruker AXS Corporation).

## Morphology of commercial PCTE membrane

Before imaging, the PCTE membrane after extrusion was rinsed with deionized water, and dried overnight at 37 °C. Then, the PCTE membrane was sputter-coated with gold and imaged by using field emission scanning electron microscopy (FE-SEM; Carl Zeiss Sigma HD | VP).

## Model and simulation of CM vesicles rupture

Simulations were implemented within a cylindrical channel with diameter $D_{chl}$ = 0.2 μm, and length $L_{sim}$ = 0.8 μm, mimicking a section of the porous media inside the PCTE membrane. Considering the flow consisting of deformable CM vesicles/liposomes and rigid NPs in the

microchannel as a three-phase flow, the simulation was conducted through the open source code Ludwig (v8)[54]. The flow field was governed by Navier-Stokes equation (with $Re \ll 1$, a typical situation for the flow in the microfluidic device) and was simulated through the Lattice Boltzmann Method. The CM vesicle/liposome was regarded as a droplet, and the deformation of which was captured by solving a modified Cahn-Hilliard equation using finite volume method, with the WENO scheme[55,56] being adopted for the discretization of the convection term to ensure the conservation of mass. With an extra couple of the rigid NPs transported by the surrounding flow, the bounce back algorithm is applied on NP surface for no-slip boundary conditions. The images were processed by using open source ParaView 5.5.2 software. The parameter settings for CM vesicle/liposome, $SiO_2$ NP, and the surrounding fluid environment were summarized in Supplementary Table 1.

### Lipid extraction
Lipids were extracted from the collected CT26 CMs using the reported methyl-tert-butyl ether (MTBE) method[57]. Briefly, 1.5 mL of cold methanol was added to the 400 μL CM suspension, and then the glass tube was vortexed. Afterward, MTBE (5 mL) was added to the mixture, and bath sonicated at 4 °C for 1 h. Next, 1.25 mL deionized water was added to produce phase separation. The mixture was centrifuged at $1000 \times g$ for 10 min. The upper organic phase was collected and transferred into a new glass tube, while the lower phase was reextracted by adding the solvent mixture (volume: MTBE/methanol/water = 10:3:2.5). Finally, the combined organic phase was evaporated under a nitrogen flow and stored at −80 °C prior to analysis.

### ESI-MS/MS analysis
Immediately before the analysis, an internal lipid standard mixture and 2% $NH_4OH$ were added to the sample. The samples were then infused from a syringe into the electrospray ionization (ESI) source of a triple quadrupole mass spectrometer (Agilent 6410 Triple Quad; Agilent Technologies, Inc., Santa Clara, USA) at a flow rate of 10 μL/min. Phospholipid species were detected selectively using head group specific MS/MS scanning modes (PC, LysoPC and SM: precursor ion 184 (P184), PE: neutral loss of 141 (NL141), PI: P241, PS: NL87). The source temperature of the ESI-MS/MS instrument was 250 °C and collision energies optimized for each lipid class (25−45 eV) were used. Nitrogen was used as the collision, nebulizing (40 psi), and drying gas (3 L/min). The obtained mass spectra were processed using MassHunter Workstation Qualitative Analysis software (Agilent Technologies, Inc., Santa Clara, USA) and the individual lipid species were quantified using the internal standards and Lipid Mass Spectrum Analysis (LIMSA) software[58], which automatically corrects for an overlap of isotope peaks.

### Differential scanning calorimetry
Phase transition temperatures of liposomes, CM vesicles, and hybrid membrane (HM) vesicles were measured using a TA Instruments Discovery differential scanning calorimetry (DSC). The samples (25 μL) were placed in the DSC sample pans, and the pans were sealed with the lids. The heating scan was started from −40 °C to 70 °C at the rate of 2 °C/min.

### Measurement of membrane fluidity
Membrane fluidity was first evaluated by generalized polarization (GP) of laurdan, as reported previously[59]. Briefly, fluorescent probe laurdan was dissolved in dimethyl sulfoxide (DMSO) to prepare the stock solution (1 mM). After incubation of liposome samples (0.1 mg/mL) in 20 mM HEPES (pH = 7.4), containing 2.5 μM of laurdan for 30 min at room temperature, the laurdan fluorescence was measured by using a multi-function microplate reader (Hidex Oy, Turku, Finland) with the excitation wavelength of 350 nm. The laurdan GP was calculated from

the emission intensities using the following equation:

$$GP = \frac{I_{440} - I_{490}}{I_{440} + I_{490}} \qquad (3)$$

where $I_{440}$ and $I_{490}$ are the fluorescence intensities measured at 440 and 490 nm, respectively.

In addition, the microviscosity of liposomes was determined by measuring steady-state fluorescence polarization of fluorescent probe 1,6-diphenyl-1,3,5-hexatriene (DPH). In brief, liposome samples (0.1 mg/mL) in 20 mM HEPES (pH = 7.4) were incubated with DPH (dissolved in tetrahydrofuran) at a final concentration of 1 μM for 60 min in the dark at room temperature. Polarization measurements were carried out using a multi-function microplate reader equipped with the polarizer (Hidex Oy, Turku, Finland). The excitation and emission wavelengths were set at 355 and 430 nm, respectively. DPH fluorescence anisotropy (r) was calculated as

$$r = \frac{I_{\parallel} - I_{\perp}}{I_{\parallel} + 2I_{\perp}} \qquad (4)$$

where $I_{\parallel}$ and $I_{\perp}$ represent the corresponding parallel and perpendicular emission fluorescence intensities with respect to the vertically polarized excitation light, respectively. The average values of microviscosity (η) were then calculated according to the Perren equation[60]:

$$\eta(cP) = \frac{240 \cdot r}{0.362 - r} \qquad (5)$$

### Preparation and characterization of HM-SiO$_2$ NPs
To obtain the HM vesicles, the lipid film as described above was hydrated with 1 mL of as-prepared CM fragments at 4 °C with occasional shaking for 1 h. The lipid suspension was then subjected to 4 freeze/thaw cycles and extruded 21 times through a 200 nm PCTE membrane. Afterwards, the HM vesicles and $SiO_2$ NPs were mixed in equal volumes, followed by extrusion through a 200 nm PCTE membrane for 21 passes to prepare the HM-$SiO_2$ NPs. To further validate the HM coating method, the polyunsaturated 1,2-dilinolenoyl-sn-glycero-3-phosphocholine (18:3 PC) was utilized to prepare HM coating under the same experimental condition. Remarkably, in a typical experiment, unless stated otherwise, the HM vesicles used for coating were made of DOPC in the following studies. For the colocalization imaging, the DOPC lipid film, CM materials and $SiO_2$ NPs were labeled with NBD-DSPE (green), DiI (orange) and Cy5 (red), respectively. HM vesicles or HM-$SiO_2$ NPs were prepared under the same experimental condition as described above using these fluorescent dye-labeled products.

The TEM imaging and DLS measurements for HM-$SiO_2$ NPs were performed following the method described above. Sodium dodecyl sulfate−polyacrylamide gel electrophoresis (SDS-PAGE) was carried out to analyze the protein composition. The colloidal stability of HM-$SiO_2$ NPs in 1X PBS were evaluated at various predetermined time periods (0, 1, 3, 6, 12, 24, 30, and 48 h) after incubation at 37 °C using DLS.

### Cell culture
CT26 cells were grown in RPMI 1640 supplemented with 10% fetal bovine serum and 1% antibiotic antimycotic solution (100×). HeLa cells, MCF-7 cells, and RAW 264.7 macrophage cells were grown in DMEM containing with 10% fetal bovine serum and 1% antibiotic antimycotic solution (100×). All of the cells were cultured in a humidified incubator with 5% $CO_2$ at 37 °C.

## Cell viability

CellTiter–Glo Luminescent Cell Viability Assay (Promega Co.) was used to evaluate the cytotoxicity of $SiO_2$ NPs, LB-$SiO_2$ NPs, CM-$SiO_2$ NPs, and HM-$SiO_2$ NPs to CT26 cells. In brief, the cells were seeded at a density of $1 \times 10^4$ cells/well in 96-well plates and cultured for 24 h. The medium was then discarded and replaced with 100 μL of fresh medium containing tested samples ($SiO_2$ NPs, LB-$SiO_2$ NPs, CM-$SiO_2$ NPs, and HM-$SiO_2$ NPs) with various concentrations (25, 50, 100, and 250 μg/mL). The cells that were not treated with NPs served as a negative control. After another 24 h incubation at 37 °C, CellTiter–Glo reagent (100 μL) was added and the contents were mixed. The plates were then incubated at room temperature for 10 min to stabilize the luminescence signals. Finally, for cell viability analysis, the luminescent intensities of the cells per well were measured using a Synergy H1 microplate reader (Biotek, Winooski, USA). Each group was replicated five times.

## Homotypic targeting studies

The in vitro homotypic targeting capacity of HM-$SiO_2$ NPs towards CT26 cells was determined using a combination of confocal laser scanning microscopy (CLSM), flow cytometry, and TEM. For CLSM imaging, CT26, HeLa, MCF-7, and RAW264.7 cells were seeded at a density of $2.5 \times 10^4$ cells per well in μ-Slide 8 well Ibidi plates (Ibidi GmbH, Germany) and incubated in 200 uL of DMEM or RPMI 1640 growth medium for 24 h. Subsequently, the growth medium was replaced with a fresh medium containing tested samples, including $SiO_2$ NPs, LB-$SiO_2$ NPs, CM-$SiO_2$ NPs, and HM-$SiO_2$ NPs with a concentration of 50 μg/mL. The cells were then orderly incubated for 4 h, washed with HBSS for three times, stained with DAPI (1 μg/mL in HBSS) for 10 min in the dark, washed again and imaged using CLSM (Zeiss LSM 800 Airyscan, Carl Zeiss, Jena, Germany). The CLSM images were obtained using the ZEN 3.1 (blue edition) software.

Cy5-labeled $SiO_2$ NPs were employed in the flow cytometric study to trace the uptake of the HM-$SiO_2$ NPs. CT26, HeLa, MCF-7, and RAW264.7 cells ($5 \times 10^5$ cells/well) were first seeded in 6-well plates and cultured in 2 mL of DMEM or RPMI 1640 medium for 24 h. After incubating the cells with tested samples ($SiO_2$ NPs, LB-$SiO_2$ NPs, CM-$SiO_2$ NPs, and HM-$SiO_2$ NPs with a concentration of 50 μg/mL) for 4 h, the cells were washed with HBSS for three times, detached by trypsin-EDTA and ultimately collected by centrifugation ($1200 \times g$, 5 min). The bottom cell pellets were washed with HBSS for three times before being examined with a BD FACSCanto II flow cytometer (BD Biosciences, San Jose, CA, USA). Collected data were analyzed using the FlowJo V10 software. Supplementary Fig. 23 shows the gating strategy used to quantify NP binding and uptake.

TEM was performed to further visualize the intracellular internalization and localization of $SiO_2$ NPs, LB-$SiO_2$ NPs, CM-$SiO_2$ NPs, and HM-$SiO_2$ NPs. Briefly, CT26 cells were cultured and collected in the same manner as indicated in flow cytometric study. After the collected cell pellets washed with HBSS for three times, glutaraldehyde (2.5%) was applied to the cells for the fixation, followed by washing with HBSS for three times. Fixed cells were embedded in 2% agarose, washed with deionized water, and then dehydrated in increasing concentrations of ethanol (i.e., 30%, 50%, 75%, 100%; 10 min for each step). Finally, the cells were processed for Epon embedding by polymerizing them for 8 h at 37 °C and 56 h at 60 °C, and then cutting them into 70–100 nm thick slices with a diamond knife. The ultrathin slices were then collected on 100-mesh copper grids, and stained with uranyl acetate (4%) for 15 min before being treated with lead citrate for 7 min. A JEM-2100F (JEM Ltd., Japan) microscope was used to image the samples.

## In vivo pharmacokinetics study

All animal experiments were carried out in accordance with the Principles of Laboratory Animal Care and approved by the Animal Ethics Committee of Anhui Medical University, China. All the animals were housed in a standard room with environmentally controlled conditions (i.e., ambient temperature of 25 °C, relative humidity of 40–60%, and lighting time at 8:00-21:00). The Sprague–Dawley (SD) rats (half male and female, weight $200 \pm 5$ g, 60–120 days) were randomly divided into four groups ($n = 3$ per group) and intravenously injected with Cy5 labeled-$SiO_2$ NPs, LB-$SiO_2$ NPs, CM-$SiO_2$ NPs, and HM-$SiO_2$ NPs ($SiO_2$ concentration: 6.25 mg/kg), respectively. At varying time-points after tail vein injection (i.e., 0.0083, 0.1666, 0.333, 0.5, 1, 2, 3, 6, 12, 19, 24, 36, and 48 h), the blood samples were collected in anticoagulant tubes from the heart and followed by centrifugation at $2350 \times g$ for 10 min to obtain plasma samples. The $SiO_2$ concentration of samples was measured using a previously reported fluorescence assay[61]. Pharmacokinetic parameters including distribution half-life ($t_{1/2\alpha}$), elimination half-life ($t_{1/2\beta}$), area under the curve (AUC), mean residence time (MRT), volume of distribution ($V_d$), and clearance rate (Cl) were determined using a two-compartment model with the DAS 2.0 software.

## In vivo targeting

To construct a subcutaneous xenograft colon tumor-bearing mouse model, the CT26 cells (around $2 \times 10^6$) were injected subcutaneously into the back of each BALB/c mouse (4–6 weeks old). When the tumor volume reached approximately 100 $mm^3$, the CT26 tumor-bearing mice were intravenously injected with Cy5 labeled-$SiO_2$ NPs, LB-$SiO_2$ NPs, CM-$SiO_2$ NPs, and HM-$SiO_2$ NPs with the identical content of $SiO_2$ NPs (100 μL, 1 mg/mL), respectively. The in vivo living imaging were performed with the IVIS imaging system (PerkinElmer, USA) at the predetermined time points (0, 4, 8, 12, 24, 48, 72, and 96 h). The treated mice were euthanized at 96 h post-injection and the tumors and major organs (heart, liver, spleen, lungs, and kidneys) were collected for the ex vivo biodistribution imaging and the hematoxylin and eosin (H&E) staining. A maximal tumor size/burden of 1200 mm3 was approved by the Animal Care Committee of Anhui Medical University and was not exceeded in our experiments.

## Statistical analysis

All the experiments including those in Supplementary Information were carried out three times independently with similar results. Results are presented as mean ± s.d. The data were analyzed for statistical significance by one-way analysis of variance (ANOVA) followed by post hoc Tukey test. Statistical analyses were performed using Origin 2019 software (OriginLabs). A $p$-value smaller than 0.05 was considered statistically significant; $**p < 0.01$, $***p < 0.001$, and $****p < 0.0001$.

## Reporting summary

Further information on research design is available in the Nature Research Reporting Summary linked to this article.

# Data availability

The authors declare that all data supporting the findings of this study are available within the paper and its supplementary information files. Source data are provided with this paper.

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

## Acknowledgements

We thank Jari Leskinen and Virpi Miettinen at SIB Labs of the University of Eastern Finland for technical support in TEM imaging. We thank Sanna P. Sihvo for the assistance of ESI–MS measurements. V.P.L. acknowledges the support from the Academy of Finland (projects 314412). D.Y.P. acknowledges the support from the National Natural Science Foundation of China (Grant Nos. 11972322 and 91852205).

## Author contributions

L.Z.L., W.J.X., and V.P.L. conceived and designed the project. L.Z.L. carried out the synthesis and performed all of the experiments except for the in vivo. D.Y.P. and S.C. performed the computational simulations. M.V.M. and M.R. assisted with the flow cytometric measurement. A.K. assisted with the SDS-PAGE characterization. J.K. carried out the analyses of pore fixing. H.P. assisted with the DSC measurement. H.R. and R.K. assisted with the ESI–MS measurement. J.W. performed the in vivo study. L.Z.L. wrote the manuscript. All authors discussed the results and commented on the manuscript.

## Competing interests

The authors declare no competing interests.
