## [Peer Review File · Nature Communications]

Systematic design of cell membrane coating to improve tumor targeting of nanoparticlesEditorial Note: This manuscript has been previously reviewed at another journal that is not operating a transparent peer review scheme. This document only contains reviewer comments and rebuttal letters for versions considered at Nature Communications.

REVIEWER COMMENTS

Reviewer #1 (Remarks to the Author):

The authors have done an impressive job in revising the manuscript to address prior comments by adding new experimental data and discussion. All prior comments have been addressed fairly well.

Reviewer #2 (Remarks to the Author):

The manuscript describes the design and preparation of nanoparticles coated with cell membranes, with the aim of improving tumor targeting of the nanoparticles. The work has a clear interest and high significance in biomedicine and bio-nanotechnology. Since most of the work done is experimental, and none of it is within my expertise, it is difficult for me to judge whether "the data presented do not support the conclusions of the manuscript", as claimed by Reviewer #1. However, there seems to be a simple way to verify if the claims of the authors are solid or not:

The main claim is that high membrane fluidity is crucial for improving coating of the NPs by cell membranes, but only one type of phospholipid membrane (made of pure DOPC) was used to increase membrane fluidity; DOPC membranes are of course fluid, but in principle it is easy to use even more fluid membranes, such as those formed by polyunsaturated lipids. In particular, using polyunsaturated PC lipids other possible contributing factors would be excluded. If membrane fluidity really is the main contributor to the improved coating of the NPs, and if adding polyunsaturated PC further improves targeting of NPs, then it would be difficult to question the conclusions of the paper.

REVIEWER COMMENTS

Reviewer #1 (Remarks to the Author):

The authors have done an impressive job in revising the manuscript to address prior comments by adding new experimental data and discussion. All prior comments have been addressed fairly well.

Response: We sincerely appreciate the reviewer's positive feedback.

Reviewer #2 (Remarks to the Author):

The manuscript describes the design and preparation of nanoparticles coated with cell membranes, with the aim of improving tumor targeting of the nanoparticles. The work has a clear interest and high significance in biomedicine and bio-nanotechnology. Since most of the work done is experimental, and none of it is within my expertise, it is difficult for me to judge whether "the data presented do not support the conclusions of the manuscript", as claimed by Reviewer #1. However, there seems to be a simple way to verify if the claims of the authors are solid or not:

The main claim is that high membrane fluidity is crucial for improving coating of the NPs by cell membranes, but only one type of phospholipid membrane (made of pure DOPC) was used to increase membrane fluidity; DOPC membranes are of course fluid, but in principle it is easy to use even more fluid membranes, such as those formed by polyunsaturated lipids. In particular, using polyunsaturated PC lipids other possible contributing factors would be excluded. If membrane fluidity really is the main contributor to the improved coating of the NPs, and if adding polyunsaturated PC further improves targeting of NPs, then it would be difficult to question the conclusions of the paper.

Response: We sincerely thank Reviewer #2 for the evaluation of our manuscript and thoughtful comments. As per your suggestion, the polyunsaturated 1,2-dilinolenoyl-sn-glycero-3-phosphocholine (18:3 PC) was used as a helper phospholipid to further validate the hybrid membrane (HM) coating method due to its higher membrane fluidity as compared to the monounsaturated DOPC (*Biochim. Biophys. Acta*, **1984**, 779, 89–137). As shown in Fig. R1, the polyunsaturated 18:3 PC used in the HM coating gave a higher ratio of full coating (~34.5%) than that coated with monounsaturated DOPC (~23.3%). Indeed, this finding further supports the conclusion that the high membrane fluidity is crucial for improving the traditional CM coating. Accordingly, we have now added the following descriptions to the revised manuscript on lines 302-308 of page 8:

"Besides the monounsaturated DOPC, the polyunsaturated 1,2-dilinolenoyl-sn-glycero-3-phosphocholine (18:3 PC) was used as a helper phospholipid to further validate the HM coating method (Supplementary Fig. 13) due to its higher membrane fluidity as compared to the monounsaturated DOPC⁴³. The polyunsaturated 18:3 PC used in the HM coating gave a higher ratio of full coating (~34.5%) than that coated with monounsaturated DOPC (~23.3%). This finding further supports the conclusion that the high membrane fluidity is crucial for improving the traditional CM coating."

The following texts in the method of "Preparation and characterization of HM-SiO₂ NPs" were added to the revised Supplementary Information (lines 188-192 of page 5):

"To further validate the HM coating method, the polyunsaturated 1,2-dilinolenoyl-sn-glycero-3-phosphocholine (18:3 PC) was utilized to prepare HM coating under the same experimental condition. Remarkably, in a typical experiment, unless stated otherwise, the HM vesicles used for coating were made of DOPC in the following studies."

Fig. R1 a, Chemical structure of polyunsaturated 1,2-dilinolenoyl-sn-glycero-3-phosphocholine (18:3 PC) used in the HM coating. **b**, Mean diameters and zeta potentials of SiO₂ NPs, HM vesicles and HM-SiO₂ NPs, in which 18:3 PC was used as a helper phospholipid to prepare HM vesicles and HM coating. **c**, TEM image of HM-SiO₂ NPs. Scale bar, 100 nm. **d**, Quantification of the ratio of full CM coating for SiO₂ NPs coated with HM vesicles and the plain CM coating with the same amount of CM protein (0.4 mg/mL) used in the HM coating. Data represent the mean \pm s.d. ($n = 3$). This figure was included as Supplementary Fig. 13 in the revised Supplementary Information.

REVIEWERS' COMMENTS

Reviewer #1 (Remarks to the Author):

The new data are sufficient to address Reviewer #2 comment.

REVIEWERS' COMMENTS

Reviewer #1 (Remarks to the Author):

The new data are sufficient to address Reviewer #2 comment.

Response: We sincerely appreciate the reviewer's positive comment.